# Impact of Preoperative CT-Measured Sarcopenia on Clinical, Pathological, and Oncological Outcomes After Elective Rectal Cancer Surgery

**DOI:** 10.3390/diagnostics15050629

**Published:** 2025-03-05

**Authors:** David Martin, Mathilde Billy, Fabio Becce, Damien Maier, Michael Schneider, Clarisse Dromain, Dieter Hahnloser, Martin Hübner, Fabian Grass

**Affiliations:** 1Department of Visceral Surgery, Lausanne University Hospital (CHUV), University of Lausanne (UNIL), 1011 Lausanne, Switzerland; david.martin@chuv.ch (D.M.); mathilde.billy@chuv.ch (M.B.); michael.schneider@chuv.ch (M.S.); dieter.hahnloser@chuv.ch (D.H.); martin.hubner@chuv.ch (M.H.); 2Department of Diagnostic and Interventional Radiology, Lausanne University Hospital (CHUV), University of Lausanne (UNIL), 1011 Lausanne, Switzerland; fabio.becce@chuv.ch (F.B.); damien.maier@chuv.ch (D.M.); clarisse.dromain@chuv.ch (C.D.)

**Keywords:** sarcopenia, computed tomography, rectal cancer, surgery, outcomes

## Abstract

**Background:** Patients with rectal cancer may be exposed to a loss of muscle strength and quality. This study aimed to assess the role of preoperative CT-based sarcopenia on postoperative clinical, pathological, and oncological outcomes after rectal cancer surgery. **Methods:** This retrospective monocentric study included patients who underwent elective oncologic resection for rectal adenocarcinoma between 01/2014 and 03/2022. The skeletal muscle index (SMI) was measured using CT at the third lumbar vertebral level, and sarcopenia was defined based on pre-established sex-specific cut-offs. Patients with sarcopenia were compared to those without sarcopenia in terms of outcomes. A Cox proportional hazard regression analysis was used to determine the independent prognostic factors of disease-free survival (DFS) and overall survival (OS). **Results:** A total of 208 patients were included, and 123 (59%) had preoperative sarcopenia. Patients with sarcopenia were significantly older (66 vs. 61 years, *p* = 0.003), had lower BMI (24 vs. 28 kg/m^2^, *p* < 0.001), and were mainly men (76 vs. 48%, *p* < 0.001). There was no difference in overall and major complication rates between the sarcopenia and non-sarcopenia group (43 vs. 37%, *p* = 0.389, and 17 vs. 17%, *p* = 1.000, respectively). Preoperative and postoperative features related to rectal surgery were comparable. The only predictive factor impacting OS was R1/R2 resection (HR 4.915, 95% CI, 1.141–11.282, *p* < 0.001), while sarcopenia (HR 2.013, 95% CI 0.972–4.173, *p* = 0.050) and T3/T4 status (HR 2.108, 95% CI 1.058–4.203, *p* = 0.034) were independently associated with DFS. **Conclusions:** A majority of patients undergoing rectal cancer surgery had preoperative CT-based sarcopenia. In this cohort, sarcopenia had no impact on postoperative morbidity and OS but was independently associated with DFS.

## 1. Introduction

Colorectal cancer is the third most common cancer and the second leading cause of cancer death globally, with rectal cancer accounting for one-third of these cases [1,2]. Rectal cancer is a complex multifactorial disease with diverse clinical presentations and various challenges related to diagnosis and treatment. Sarcopenia refers to a condition characterized by the progressive loss of skeletal muscle mass and strength, typically observed with advancing age and also associated with various chronic health conditions [3]. Gradual decline in skeletal muscle mass and function has become a significant health concern in recent years and has been associated with increased adverse outcomes including falls, functional decline, frailty, and mortality [4,5,6,7,8]. Although it is primarily a disease of the elderly, its development may be associated with conditions such as cancer. Sarcopenia has therefore drawn considerable attention to oncologic research in many gastrointestinal cancers [9,10,11].

On a practical level, an approach suggested by the European Working Group on Sarcopenia in Older People (EWGSOP) is to measure the areas of specific muscle groups through imaging and to assess muscle quantity and quality [12]. While this provides quantitative and qualitative measures, functional status is not assessed. Most cancer patients require cross-sectional imaging, and these parameters are therefore easily measurable and accessible, compared to more complex functional evaluation. The analysis of cross-sectional CT images at the level of the third lumbar vertebra allows the assessment of paravertebral and abdominal wall muscles and is recognized by the EWGSOP consensus to identify sarcopenia based on imaging [12]. Several body composition indices can be measured on these CT datasets, including skeletal muscle area (SMA), muscle density (skeletal muscle radiation attenuation (SMRA), and fat infiltration (intermuscular adipose tissue content, IMAT).

In rectal cancer, tumor stage or lymph node involvement represent important prognostic factors strongly associated with the disease course [13]. Nevertheless, recent studies have recognized the role of sarcopenia as a potential prognostic marker for poor clinical and oncological outcomes in patients with rectal cancer [14,15,16,17,18]. However, the definitions of sarcopenia were not consistent. Furthermore, several studies also included patients with colon cancer. In this context, there is a need to further confirm the relationship between sarcopenia and outcomes in patients with rectal cancer.

The aim of the present study was to assess the presence of CT-based sarcopenia in patients undergoing elective rectal cancer surgery and to assess its potential impact on postoperative clinical, pathological, and oncological outcomes.

## 2. Materials and Methods

### 2.1. Study Design and Patients

This retrospective monocentric study included consecutive patients who underwent elective oncologic resection for rectal adenocarcinoma within 15 cm from the anal verge at the Department of Visceral Surgery, Lausanne University Hospital (CHUV), Switzerland, between January 2014 and March 2022. All patients were treated according to the Enhanced Recovery After Surgery (ERAS) protocol for rectal cancer surgery [19]. Treatment strategies were discussed preoperatively in the setting of multidisciplinary tumor board meetings including the need and type of neoadjuvant treatment, according to European guidelines [20]. Different combinations of short- and long-course radiotherapy and either induction or consolidation chemotherapy regimens were used in the setting of this study and summarized as neoadjuvant treatment. Patients with sarcopenia were compared to those without sarcopenia in terms of demographics, surgical details, postoperative morbidity, mortality, and survival.

### 2.2. Variables of Interest and Outcomes

Patients’ demographics, comorbidities, tumor location, and surgical details were collected. Postoperative outcomes included morbidity, mortality, reoperations, length of hospital stay (LOS), pathological analyses, and survival. Complications were assessed according to the Clavien–Dindo classification within 30 days of surgery [21]. Major complications included Clavien–Dindo grade ≥ IIIa. The highest grade was considered in patients presenting several complications. The comprehensive complication index (CCI) was computed to reflect overall morbidity, with a score ranging from 0 to 100 [22]. Specific preoperative and postoperative features related to rectal surgery, including the circumferential resection margin (CRM), were assessed. Anastomotic leaks were confirmed radiologically with the administration of intrarectal contrast material. The tumor stage was assessed with pelvic MRI and described according to the Union for International Cancer Control (UICC) TNM classification [23]. The CRM was defined as involved (positive) if the tumor-free margin was ≤1 mm [24].

### 2.3. Assessment of Body Composition and Sarcopenia

All CT scans were performed preoperatively within a time interval of 3 months before surgery, with or without intravenous contrast administration. Body composition indices were measured on a single axial CT slice at the third lumbar vertebral level using a semi-automated deep-learning-based method with a U-Net architecture algorithm, as previously described [25,26]. These muscle segmentations were all reviewed and corrected by an experienced (14 years) musculoskeletal radiologist and included the psoas muscles, the paraspinal muscles, and the abdominal wall muscles (Figure 1).

Muscle mass (quantity) was represented by the SMA (in cm^2^), while muscle quality was reflected by the SMRA (in Hounsfield unit, HU) and IMAT (in cm^2^). The SMA and IMAT were, respectively, normalized for patient height to obtain the skeletal muscle index (SMI) and IMAT index (IMATI), expressed in cm^2^/m^2^. The definition used to establish the presence of sarcopenia was an SMI cut-off ≤ 38.5 cm^2^/m^2^ in women and ≤52.4 cm^2^/m^2^ in men, as previously described [27,28].

### 2.4. Statistical Analysis

Continuous variables were expressed as mean (standard deviation, SD) or median (interquartile range, IQR) and compared using the Mann–Whitney U test or Student’s *t*-test, according to their distribution. Categorical variables were displayed as frequencies (%) and compared among groups using Pearson’s chi-squared or Fisher’s exact test, as appropriate. Overall survival (OS) and disease-free survival (DFS) were assessed with the Kaplan–Meier method, and groups were compared using the log-rank test. OS and DFS were defined as the time interval between the day of surgery and the date of death due to any cause or tumor recurrence, respectively. Relevant clinicopathological characteristics were evaluated using univariable analysis of their association with OS and DFS. Variables with a *p*-value  ≤ 0.1 were included for multivariable Cox proportional hazard regression analysis. Results were reported as hazard ratios (HRs) with corresponding 95% confidence intervals. A *p*-value < 0.05 was considered to indicate statistical significance. All analyses were performed using SPSS 28.0 software (SPSS Inc., Chicago, IL, USA).

## 3. Results

A total of 208 patients were included, and 123 (59%) had preoperative CT-based sarcopenia. Patient demographics and surgical details are presented in Table 1. The mean age was 64 years, and 134 (64%) were men. Patients with sarcopenia were significantly older (66 vs. 61 years, *p* = 0.003), had lower BMI (24 vs. 28 kg/m^2^, *p* < 0.001), and were mainly men (76 vs. 48%, *p* < 0.001). Sarcopenic patients had significantly higher ASA scores ≥ III (39 vs. 24%, *p* = 0.024). The two groups were comparable for the remaining comorbidities, neoadjuvant treatments, tumor location, and surgical details.

The median LOS was 9 days (IQR 5–14) in the sarcopenia group and 6 days (IQR 5–11) in the non-sarcopenia group (*p* = 0.092, Table 2). The overall complication rate was 43% (*n* = 53) in the sarcopenic group and 37% (*n* = 31) in the non-sarcopenic group (*p* = 0.389). There was no difference in major complication rates between the sarcopenia and non-sarcopenia groups (17% vs. 17%, *p* = 1.000). Mortality, readmission, and reoperation rates were comparable between the two groups. There were no differences in anastomotic leak rates (5 vs. 7%, *p* = 0.554), and CRM rates were comparable between the two groups. Tumor stage and nodal involvement were comparable, while sarcopenic patients underwent significantly more R1/R2 resections and presented with higher distant recurrence rates than non-sarcopenic patients (20 vs. 8%, *p* = 0.025, and 26 vs. 8%, *p* = 0.002, respectively).

The median OS of patients with sarcopenia was comparable to those without sarcopenia (37.3 vs. 37.0 months, *p* = 0.312, Figure 2). The median DFS was significantly lower in patients with sarcopenia than in patients without sarcopenia (25.9 vs. 32.5 months, *p* = 0.002).

In the Cox proportional hazard regression analysis, the only significant predictive factor for OS was R1/R2 resection (HR 4.915, 95% CI, 1.141–11.282, *p* < 0.001, Table 3). Sarcopenia (HR 2.013, 95% CI 0.972–4.173, *p* = 0.050) and T3/T4 status (HR 2.108, 95% CI 1.058–4.203, *p* = 0.034) were independently associated with DFS (Table 4). Sarcopenia was the only factor associated with distant recurrence (HR 2.989, 95% CI 1.309–6.826, *p* = 0.009, Appendix A). No factor influencing local recurrence was identified (Appendix A).

## 4. Discussion

In this cohort, nearly two-thirds of patients undergoing rectal cancer surgery had preoperative CT-based sarcopenia. Sarcopenia had no impact on early postoperative morbidity, mortality, length of stay, or OS. However, sarcopenia was independently associated with DFS and distant recurrence.

In a recent systematic review, the global prevalence of sarcopenia ranged from 5% to 17% among the elderly [29]. Similarly to the present study, a meta-analysis showed the prevalence of sarcopenia in the rectal cancer population ranging from 25 to 68% [14]. Several different definitions of sarcopenia were used, and the quality of the evidence was rated as moderate, thus explaining the wide prevalence range. Four of the seven studies applied the same sarcopenia cut-offs as this present study and were based on the SMI measured at the third lumbar vertebra level on CT scans. This imaging modality is routinely available and convenient for easily assessing body composition without the need for additional modalities since it is part of the preoperative standard work-up for rectal cancer. However, the cut-off values are neither precisely defined nor validated, both for muscle quantity and quality. In the present study, pre-established sex-specific SMI cut-offs based on previous studies were used. This may introduce bias given a potential under- or overestimation of sarcopenia depending on race, ethnicity, and further unknown factors. An American study revealed a discrepancy among Hispanics, Asians, Blacks, and Whites across sarcopenia indices, suggesting differences in muscle mass/quality or functional changes with aging [30]. In addition, the incidence of sarcopenia varies according to the definitions used. In the vast majority of current rectal cancer studies, as in this present study, the diagnosis of sarcopenia is based on SMI assessment alone, but not muscle strength and physical performance, in line with recommendations of the EWGSOP consensus [12]. This is probably due to the fact that imaging modalities are available at no additional costs, resources, or time as opposed to functional measures such as strength grip or gait tests, for example. This is a limitation to consider when interpreting and generalizing the results.

Sarcopenic patients were significantly older, had a lower BMI, and had higher ASA scores compared to non-sarcopenic patients in the present study. These observations are consistent with previous studies [31,32,33,34]. Two retrospective studies on rectal cancer further revealed a preponderance of sarcopenia in male patients (62.2–70.8%) with lower BMI compared to non-sarcopenic groups [35,36]. Potential explanations for male preponderance are related to hormonal variations and declining testosterone levels with age, leading to decreased muscle mass and strength [37]. It also appears that sarcopenic patients tend to have BMIs within normal ranges. On the other hand, obesity, and more specifically visceral obesity, have been identified as poor prognostic factors in colorectal cancer [38,39]. The term “sarcopenic obesity” has recently spread and is defined as the presence of both high body fat percentage and sarcopenia. Hence, sarcopenic obesity should be considered a unique clinical condition, different from obesity or sarcopenia alone [40]. While these parameters were not assessed in the present study, a recent retrospective study on locally advanced rectal cancer patients found that sarcopenic obesity was associated with both worse DFS and OS [41].

The decline of skeletal muscle mass and quality in rectal cancer patients could be explained by complex interactions between patient characteristics (nutritional status, age, sex, comorbidities), unfavorable tumor biology, and chemoradiotherapy. A previous retrospective study including colorectal cancer patients showed that data derived from CT images presented a large variation in muscle quantity and quality (SMI, SMRA, and IMAT) [42]. In the present cohort, muscle quantity (SMI, SMA) was a predictor of poorer oncological outcomes (DFS and distant recurrence), while muscle quality (SMRA, IMATI) was not. Indeed, muscle quantity does not necessarily correlate with muscle quality. In fact, only a few studies focused on the impact of muscle quality and quantity in rectal cancer surgery. A previous study including 308 colorectal cancer patients confirmed that a preoperative decline in muscle quantity was a more reliable indicator of poorer postoperative DFS and OS compared to muscle quality [43].

A meta-analysis including 18,891 colorectal cancer patients showed that sarcopenia was a predictor of postoperative complications [44]. Sarcopenic patients presented with more severe postoperative complications and prolonged length of stay, while anastomotic leak rates were similar. In the present study, overall and major complication rates were comparable between both groups, as was LOS. In line with our findings, two retrospective studies exclusively focusing on rectal cancer showed no impact of sarcopenia on postoperative complications [34,35]. Furthermore, the anastomotic leak rate in the present study was not higher in the sarcopenic group, similar to the findings of a recent meta-analysis including 12 studies and 5337 colorectal cancer patients [15]. These conflicting results regarding complications may be related to different definitions of sarcopenia and type II error due to low sample size.

Two meta-analyses identified sarcopenia as a negative predictor of OS and DFS in patients with resected non-metastatic colorectal cancer [15,44]. Interestingly, the present study found an association of sarcopenia with DFS and distant recurrence, but not OS. More specifically to rectal cancer, a meta-analysis involving seven studies and 2377 patients showed that sarcopenia assessed with the SMI at the third lumbar vertebra level was an independent predictor of OS (HR 2.37, 95% CI 1.13–4.98, *p* = 0.020, (14)). The quality of evidence for OS was considered moderate, and DFS was not assessed. Furthermore, the cohorts included were heterogeneous, with small collectives, and did not exclusively involve patients undergoing rectal surgery.

The reason for the negative association of sarcopenia and OS after rectal cancer surgery remains unclear. Patients with cancer are at high risk of malnutrition, and the cancer itself releases inflammatory cytokines that affect brain, muscle, liver, and fat function [34]. These phenomena may induce an anabolic–catabolic imbalance leading to muscle wasting, which reduces muscle mass and strength. However, the direct causal relationship between preoperative sarcopenia and oncological outcomes after rectal cancer surgery is not clearly established and is complex to study. Further investigations are needed to uncover the potential molecular mechanisms driving muscle loss. Epidemiological studies have shown that physical activity protects against cancer recurrence [45]. A randomized controlled trial in patients undergoing major abdominal surgery suggested that resistance exercise training may help modify muscle mass and improve postoperative outcomes [46]. Thus, preoperative imaging-based screening for sarcopenia followed by interventions and training, such as prehabilitation programs, should be considered in the future. However, prehabilitation may encounter logistic and patient-related difficulties, and it remains to define their modalities (nutritional, physical, psychological) and frequencies of application [47].

Interestingly, an association between sarcopenic patients and distant recurrence was identified by the present study, while local recurrence rates were similar. A similar association was found in a retrospective study that identified sarcopenia as a risk factor; however, for both postoperative local and distant recurrence in patients with non-small-cell lung cancer [48]. It has been suggested that the metastatic process could be facilitated by the sarcopenia-related compromised immune function and impaired general status, which could promote the dissemination and survival of cancer cells in the circulation and thus facilitate their distant spread. The cancer microenvironment also plays a crucial role in cancer progression and recurrence. A sarcopenia-related increase in systemic inflammation may promote a microenvironment favorable for the spread of distant cancer cells [49].

The only significant factor of importance to OS in the present study was R1/R2 resection, which was higher in the sarcopenia group (20 vs. 8%). The results thus suggest that sarcopenia may be associated with a more challenging surgical resection and potentially compromised clear margins. One hypothesis may be related to decreased tissue integrity, which contributes to a more challenging surgery, leading to poorer pathological specimens. However, this association has not been described previously, and further research is warranted.

This study has several limitations related, but not limited to, the monocentric and retrospective design. The rather small sample size may induce type II bias and impede the generalization of the results. Patients had a CT within 3 months before surgery. This represents an important time lapse, with potential variations in body composition measures. Furthermore, sarcopenia was measured at one specific time point. However, its dynamic evolution over time may even more closely correlate with outcomes. It is important to keep in mind that the differences in ethnicity leading to variation in muscle mass and function may yield different results. Nevertheless, a consensus on optimal cut-offs to define sarcopenia is still lacking, and this should be investigated in the future with CT data from healthy patients (e.g., living kidney donors, young trauma patients). Functional status, which is required for the definitive diagnosis of sarcopenia, was not assessed. Finally, the complex treatment of locally advanced rectal cancer includes different neoadjuvant treatment strategies, which were individualized to the specific patient and disease presentation in the setting of this study. Neoadjuvant treatment (both chemo- and radiotherapy) may have a significant impact on both sarcopenia and postoperative clinical and oncological outcomes. However, the multitude of neoadjuvant treatment strategies and the rather small sample size impeded further analysis.

Future, well-designed studies overcoming the above-mentioned limitations related to the design of the present study with larger sample sizes are needed to further assess the impact and value of sarcopenia measurements on postoperative short- and long-term outcomes.

## 5. Conclusions

A majority of patients who underwent rectal cancer surgery had preoperative CT-based sarcopenia. Sarcopenia had no significant impact on postoperative clinical outcomes including morbidity, mortality, length of stay, and OS but was independently associated with DFS due to increased distant recurrence. Based on these findings, the routine assessment of CT-based sarcopenia for prognostic purposes or preoperative interventions cannot be advocated in rectal cancer surgery.

## Figures and Tables

**Figure 1 diagnostics-15-00629-f001:**
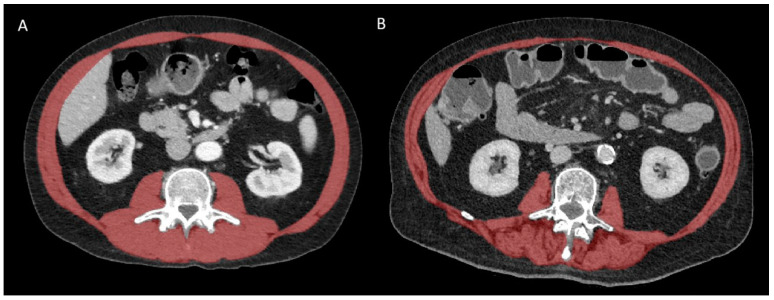
Representative CT images at L3 vertebral level in patients with (**A**) high and (**B**) low muscle mass and quality. The global skeletal muscle area (SMA) is highlighted in red before applying thresholding (−29 to +150 HU) to extract the intermuscular adipose tissue (IMAT) area. The skeletal muscle radiation attenuation (SMRA) is the average density of all pixels within SMA that represent skeletal muscle itself after thresholding.

**Figure 2 diagnostics-15-00629-f002:**
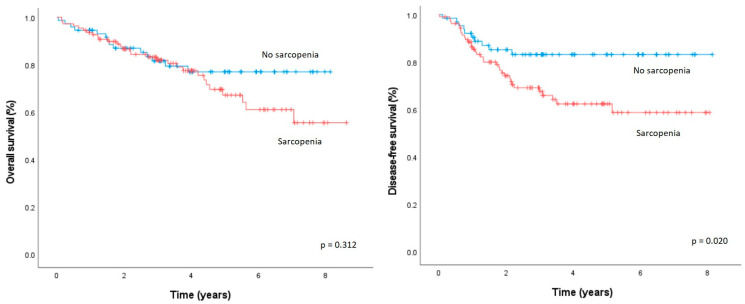
Overall survival and disease-free survival after elective oncologic rectal cancer resection, according to the presence/absence of sarcopenia.

**Table 1 diagnostics-15-00629-t001:** Demographics and surgical details.

	Overall (*n* = 208)	Non-Sarcopenia (*n* = 85)	Sarcopenia (*n* = 123)	*p*-Value *
Age (years) (mean, SD)	64 (13)	61 (12)	66 (14)	**0.003**
BMI (kg/m^2^) (mean, SD)	26 (5)	28 (5)	24 (4)	**<0.001**
Sex (male), *n* (%)	134 (64)	48 (48)	93 (76)	**<0.001**
ASA score ≥ III, *n* (%)	68 (33)	20 (24)	48 (39)	**0.024**
SMA (cm^2^) (mean, SD)	132 (34)	147 (35)	121 (29)	**<0.001**
SMI (cm^2^/m^2^) (mean, SD)	45 (10)	40 (7)	53 (9)	**<0.001**
SMRA (HU) (mean, SD)	38 (9)	39 (8)	37 (9)	0.078
IMAT (cm^2^) (mean, SD)	16 (11)	16 (9)	16 (12)	0.915
IMATI (cm^2^/m^2^) (mean, SD)	5 (4)	6 (3)	5 (4)	0.463
Cardiovascular disease, *n* (%)	43 (21)	15 (18)	28 (23)	0.390
Chronic pulmonary disease, *n* (%)	15 (7)	5 (6)	10 (8)	0.597
Diabetes, *n* (%)	21(10)	11 (12)	11 (9)	0.507
Neoadjuvant treatment, *n* (%)	138 (66)	60 (71)	78 (63)	0.300
Reverse treatment, *n* (%)	28 (14)	13 (15)	15 (12)	0.520
Rectal tumor location **, *n* (%)				0.987
Low	78 (37,5)	32 (38)	46 (38)	
Mid	82 (39)	33 (39)	49 (40)	
Upper	48 (23)	20 (24)	28 (23)	
CRM ***, *n* (%)				0.089
Negative	157 (91)	67 (96)	90 (88)	
Positive	15 (9)	3 (4)	12 (12)	
Initial operative approach, *n* (%)				0.060
Laparoscopy	130 (63)	47 (55)	83 (68)	
Open	22 (11)	8 (9)	14 (12)	
Robotic	55 (27)	30 (35)	25 (21)	
Conversion rate ****, *n* (%)	33 (17)	12 (15)	21 (19)	0.563
Protective ostomy, *n* (%)	141 (68)	61 (72)	80 (65)	0.366

* Significant *p*-values (<0.05) are displayed in bold characters/** rectal tumor location: low (0–5 cm), mid (5–10 cm), upper (10–15 cm)/*** CRM: Circumferential resection margin, 36 missing data. The percentages are adapted/**** Percentage calculated among laparoscopic and robotic cases (*n* = 196)/SD: standard deviation; BMI: body mass index; ASA: American Society of Anesthesiologists; SMA: skeletal muscle area; SMI: skeletal muscle index; SMRA: skeletal muscle radiation attenuation; HU: Hounsfield unit; IMAT: intermuscular adipose tissue; IMATI: intermuscular adipose tissue index; CRM: circumferential resection margin.

**Table 2 diagnostics-15-00629-t002:** Clinical and pathological postoperative outcomes.

Outcome	Overall (*n* = 208)	Non-Sarcopenia (*n* = 85)	Sarcopenia (*n* = 123)	*p*-Value *
Length of stay, median [IQR]	8 (5–12)	6 (5–11)	9 (5–14)	0.092
Overall complications, *n* (%)	84 (40)	31 (37)	53 (43)	0.389
Major **	35 (17)	14 (17)	21 (17)	1.000
CCI, mean (SD)	14 (22)	13 (24)	15 (20)	0.547
Reoperation, *n* (%)	23 (11)	8 (9)	15 (12)	0.655
Mortality, *n* (%)	2 (1)	1 (1)	1 (1)	1.000
Readmissions, *n* (%)	28 (14)	14 (17)	14 (11)	0.308
Anastomotic insufficiency, *n* (%)	12 (6)	6 (7)	6 (5)	0.554
T status, *n* (%)				0.403
T0 ***	19 (9)	8 (9)	11 (9)	
Tis	4 (2)	2 (2)	2 (2)	
T1–T2	60 (29)	31 (37)	29 (24)	
T3–T4	125 (60)	44 (52)	81 (66)	
N status, *n* (%)				0.216
N0	130 (62)	52 (61)	78 (63)	
N1/N2	78 (38)	33 (39)	45 (37)	
Resection status, *n* (%)				**0.025**
R0	177 (85)	78 (92)	100 (80)	
R1–R2	31 (15)	7 (8)	25 (20)	
Recurrence, *n* (%)				
Local	12 (6)	3 (4)	9 (7)	0.262
Distant	39 (19)	7 (8)	32 (26)	**0.002**

* Significant *p*-values (<0.05) are displayed in bold characters/** Major complications are defined with a Clavien grade ≥ IIIa *** T0: no residual carcinoma cells found at pathology. The percentages are adapted. CCI: comprehensive complication index.

**Table 3 diagnostics-15-00629-t003:** Cox proportional hazard regression analysis of factors influencing overall survival.

	Univariate Analysis		Multivariate Analysis	
	HR (95% CI)	*p*-Value *	HR (95% CI)	*p*-Value *
Age	1.031 (1.005–1.058)	**0.021**	1.027 (0.998–1.057)	0.070
BMI	0.981 (0.921–1.045)	0.554		
ASA ≥ 3	3.167 (1.733–5.787)	**<0.001**	2.034 (0.990–4.181)	0.053
Sarcopenia	1.427 (0.754–2.702)	0.274		
SMRA	0.971 (0.939–1.005)	0.091	0.984 (0.946–1.024)	0.431
IMATI	1.002 (0.449–2.239)	0.996		
CRM involvement	2.082 (0.870–4.983)	0.099	0.410 (0.141–1.192)	0.102
Neoadjuvant treatment	1.468 (0.753–2.862)	0.259		
Postoperative complications	2.219 (1.210–4.070)	**0.010**	1.352 (0.664–2.751)	0.406
T3/T4 status	2.454 (1.208–4.983)	**0.013**	1.352 (0.664–2.751)	0.315
N1/2 status	1.595 (0.874–2.911)	0.128		
R1/R2 resection	4.050 (2.176–7.538)	**<0.001**	4.915 (1.141–11.282)	**<0.001**

* Significant *p*-values (<0.05) are displayed in bold characters. Variables with a *p*-value  ≤  0.1 were candidates for multivariable analysis. HR: hazard ratio; BMI: body mass index; ASA: American Society of Anesthesiologists; SMRA: skeletal muscle radiation attenuation; IMATI: intermuscular adipose tissue index; CRM: circumferential resection margin.

**Table 4 diagnostics-15-00629-t004:** Cox proportional hazard regression analysis of factors influencing disease-free survival.

	Univariate Analysis		Multivariate Analysis	
	HR (95% CI)	*p*-Value *	HR (95% CI)	*p*-Value *
Age	1.011 (0.988–1.035)	0.353		
BMI	1.004 (0.952–1.060)	0.874		
ASA ≥ 3	1.891 (1.038–3.445)	**0.037**	1.324 (0.707–2.480)	0.380
Sarcopenia	2.496 (1.233–5.054)	**0.011**	2.013 (0.972–4.173)	**0.050**
SMRA	0.959 (0.928–0.992)	**0.034**	0.976 (0.930–1.023)	0.213
IMATI	1.064 (1.002–1.131)	0.054	1.001 (0.918–1.093)	0.773
CRM involvement	0.609 (0.147–2.526)	0.494		
Neoadjuvant treatment	1.565 (0.806–3.040)	0.186		
Postoperative complications	2.048 (1.130–3.711)	**0.018**	1.715 (0.930–3.162)	0.084
T3/T4 status	2.386 (1.203–4.734)	**0.013**	2.108 (1.058–4.203)	**0.034**
N1/2 status	1.512 (0.834–2.740)	0.173		
R1/R2 resection	1.068 (0.451–2.528)	0.882		

* Significant *p*-values (<0.05) are displayed in bold characters. Variables with a *p*-value  ≤  0.1 were candidates for multivariable analysis. HR: hazard ratio; BMI: body mass index; ASA: American Society of Anesthesiologists; SMRA: skeletal muscle radiation attenuation; IMATI: intermuscular adipose tissue index; CRM: circumferential resection margin.

## Data Availability

The data published in this research are available on request from the first and last author and corresponding author.

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
