# Peer review of "Impact of Preoperative CT-Measured Sarcopenia on Clinical, Pathological, and Oncological Outcomes After Elective Rectal Cancer Surgery"

_diagnostics, 2025, doi:10.3390/diagnostics15050629_

Round 1
Reviewer 1 Report
Comments and Suggestions for Authors
1. In the Materials and methods section of this study (lines 82-84), the authors refer to a comprehensive treatment approach for the included cases, based on European guidelines: "Treatment strategies were discussed preoperatively in the setting of multidisciplinary tumor board meetings including need and type of neoadjuvant treatment, according to European guidelines".
In the Materials and methods section of this study, the authors refer to a comprehensive treatment approach for the included cases, based on European guidelines, but the article does not mention a detailed description of which patients underwent treatment other than surgery, such as radiotherapy or chemotherapy.
In principle, preoperative or postoperative chemoradiotherapy will have a high probability of affecting clinical outcomes such as OS and DFS, and even preoperative treatment will affect the SMI of L3.
Please explain the above questions in detail, and if necessary, please supplement the data or explain in limitation.
2. For the segmentation and calculation of SMI parameters of L3 mentioned in the study, whether a single tomography image or multiple (3-5 layers) are used, please further explain.
3. In Figure1, the description of Muscles included in L3 lumbar vertebrae Skeletal Muscle Area (SMA) is not comprehensive, please supplement.
Author Response
Comment 1: 1. In the Materials and methods section of this study (lines 82-84), the authors refer to a comprehensive treatment approach for the included cases, based on European guidelines: "Treatment strategies were discussed preoperatively in the setting of multidisciplinary tumor board meetings including need and type of neoadjuvant treatment, according to European guidelines", but the article does not mention a detailed description of which patients underwent treatment other than surgery, such as radiotherapy or chemotherapy. In principle, preoperative or postoperative chemoradiotherapy will have a high probability of affecting clinical outcomes such as OS and DFS, and even preoperative treatment will affect the SMI of L3.
Please explain the above questions in detail, and if necessary, please supplement the data or explain in limitation.
Reply 1: Thank you for this important comment and the opportunity to specify. Indeed, neoadjuvant treatment protocols evolved during the almost a decade long study period. Over the inclusion period, we observed a shift away from the classic 5-week chemoradiation (CRT) for locally advanced rectal cancer to a more individual approach including total neoadjuvant treatment (TNT) strategies, either induction chemotherapy followed by long-course CRT, or short-course radiotherapy followed by consolidation chemotherapy. In case of a complete clinical response (which can be expected in up to 25% of patients), patients were followed through a watch-and-wait strategy and hence not included in the study. The rather small sample size impeded subgroup analysis of this heterogeneous treatment protocols, and we decided to regroup patients with neoadjuvant treatment together. This represents indeed a limitation of the study.
These considerations were added to the manuscript.
Comment 2: 2. For the segmentation and calculation of SMI parameters of L3 mentioned in the study, whether a single tomography image or multiple (3-5 layers) are used, please further explain.
Reply 2: It was a single axial CT slice .This was specified in the respective paragraph, together with further precisions regarding calculations and assessment (page 3, lines 107-111).
Comment 3: In Figure1, the description of Muscles included in L3 lumbar vertebrae Skeletal Muscle Area (SMA) is not comprehensive, please supplement.
Reply 1: Thank you for pointing this out and the opportunity to amend the figure accordingly.
We would like to thank the reviewer for the thorough assessment of our manuscript and the valuable input.
Reviewer 2 Report
Comments and Suggestions for Authors
Dear authors
This is very interesting topic of research and a well-written manuscript.
I have a few remarks about your paper:
The Introduction section is correct and I have no special remarks; please check minor spelling errors (i.e. skeletal muscle "aera" from row 62);
In subsection 2.3 "Assessment of body composition and sarcopenia", I would recommend inserting the formulas for various parameters like skeletal muscle index and skeletal muscle area, as well other parameters that you used;
Move figure 1 next to Materials and Methods, where it is mentioned; also, please notice that it seems that some text is missing from its legend; I would also recommend splitting the image into two subimages, 1A and 1B, and describe each of them separately;
In table 1, you mentioned that upper rectum is between 10-15 cm, but in Materials and Methods you specify (row 79) that you selected patients with tumors up to 12 from the anal verge; please address this inconsistency;
I don't see where table 3 is referenced in the main text.
You could expand the written text where you describe your results. Don't address only the results that are statistical significant, mention also other associations that you consider relevant, even if the association was not statistical significant;
In the final part of Discussion, insert a paragraph about Future Research Directions related to this topic.
The Conclusions are too brief; try to expand them, mentioning the main results from your study;
Please notice that references 32 and 47 could be considered as autocitations.
Author Response
Comment 1: The Introduction section is correct and I have no special remarks; please check minor spelling errors (i.e. skeletal muscle "aera" from row 62);
Thank you for pointing this out, spelling errors were corrected throughout the manuscript.
Comment 2: In subsection 2.3 "Assessment of body composition and sarcopenia", I would recommend inserting the formulas for various parameters like skeletal muscle index and skeletal muscle area, as well other parameters that you used.
Response 2: The paragraph was amended accordingly.
Comment 3: Move figure 1 next to Materials and Methods, where it is mentioned; also, please notice that it seems that some text is missing from its legend; I would also recommend splitting the image into two subimages, 1A and 1B, and describe each of them separately
Response 3: The figure and figure legend were amended accordingly. We thank the EO to display figures and legends in the text according to journal style.
Comment 4:
In table 1, you mentioned that upper rectum is between 10-15 cm, but in Materials and Methods you specify (row 79) that you selected patients with tumors up to 12 from the anal verge; please address this inconsistency
Response 4: We apologize for the confusion: 10-15 cm was indeed considered as upper rectum for the purpose of this study and the methods sections was corrected accordingly.
Comment 5: I don't see where table 3 is referenced in the main text.
Response 5: We added the missing reference of Table 3 and thank the EO to insert tables where deemed appropriate within the text. Table 3 is referenced at page 5 line 171.
Comment 6: You could expand the written text where you describe your results. Don't address only the results that are statistical significant, mention also other associations that you consider relevant, even if the association was not statistical significant
Response 6: We appreciate the suggestion. For the sake of brevity, our routine is not to duplicate information in tables and results to highlight either very important or significant results and kindly refer to the table for further associations. We prefer to discuss the associations in the discussion section if of clinical relevance.
Comment 7: In the final part of Discussion, insert a paragraph about Future Research Directions related to this topic.
Reply 7: We gladly added a paragraph as suggested.
Comment 8: The Conclusions are too brief; try to expand them, mentioning the main results from your study.
Reply 8: The conclusion section was amended as suggested with a brief summary of the main results of the study.
Comment 9: Please notice that references 32 and 47 could be considered as autocitations.
Reply 9: Thank you for pointing this out. Our group has indeed a particular interest in preoperative optimization strategies, and included earlier institutional experience considering it they are pertinent to contectualize the present study. By no means it is our intent to autocite a maximum of our own work, with 2 institutional out of 49 references representing a rather low percentage of autocitations.
We would like to thank the reviewer for the valuable comments, which helped us to improve our manuscript.
Round 2
Reviewer 1 Report
Comments and Suggestions for Authors
Most raised issues have been addressed in the revised manuscript, and can be accepted.
Reviewer 2 Report
Comments and Suggestions for Authors
Dear authors
Thank you for the extensive revisions you have made to the manuscript, according to the suggestions.
I have no further remarks.